# Position Regression for Unsupervised Anomaly Detection

**Florentin Bieder**[1]                                    FLORENTIN.BIEDER@UNIBAS.CH

**Julia Wolleb**[1]                                          JULIA.WOLLEB@UNIBAS.CH

**Robin Sandkühler**[1]                             ROBIN.SANDKUEHLER@UNIBAS.CH

**Philippe C. Cattin**[1]                              PHILIPPE.CATTIN@UNIBAS.CH

[1] *Department of Biomedical Engineering, University of Basel*

## Abstract

In recent years, anomaly detection has become an essential field in medical image analysis. Most current anomaly detection methods for medical images are based on image reconstruction. In this work, we propose a novel anomaly detection approach based on coordinate regression. Our method estimates the position of patches within a volume, and is trained only on data of healthy subjects. During inference, we can detect and localize anomalies by considering the error of the position estimate of a given patch. We apply our method to 3D CT volumes and evaluate it on patients with intracranial haemorrhages and cranial fractures. The results show that our method performs well in detecting these anomalies. Furthermore, we show that our method requires less memory than comparable approaches that involve image reconstruction. This is highly relevant for processing large 3D volumes, for instance, CT or MRI scans.

**Keywords:** medical image analysis, anomaly detection, unsupervised

## 1. Introduction

In recent years, anomaly detection has become an essential direction of research in medical image analysis. Compared to supervised segmentation methods, anomaly detection methods do not rely on pixel-wise annotations but on image-level labels instead. This leads to a much simpler way of annotating the training data and reduces the human bias to the model.

We can distinguish two major types of anomaly detection methods in the literature: The first type only uses data that is considered normal. In terms of medical images, these are images of healthy subjects. The second type additionally requires examples of anomalous data (Battikh and Lenskiy, 2021; Wolleb et al., 2020), and can also be considered as weakly supervised methods. However, in this work we will focus on the first type that uses only normal data for the training.

The most widely used methods for image anomaly detection are based on reconstruction errors (Baur et al., 2021). These methods aim to capture the distribution of the training set of healthy subjects by learning a low dimensional representation of the input and the reconstruction from this representation back to the original image. The core idea is that the correct reconstruction of the input will fail in some regions if an anomaly is present in the input image. Comparing the output with the input will provide a reconstruction error in the pixel space, which can be used as an indicator for anomalies (Chen and Konukoglu, 2018; M.D. et al., 2018; Baur et al., 2021). A challenge in these methods is generating an output image of high quality and rich in detail. This requirement contributes to the

computational cost in terms of training time and hardware requirements, i.e., GPUs and memory. (Tong et al., 2021) have also observed that autoencoders can have a bias towards data that can easily be reconstructed and are sensitive to outliers in the training set.

This paper presents a novel anomaly detection method for medical images based on position regression. In contrast to the image reconstruction-based methods our approach operates on patches of the input image. Instead of learning a reconstruction of the patch in pixel space, our method predicts the location of the input patch in the original image. Our method is trained only on data of healthy subjects and implicitly learns the distribution of the data as a result of the position regression task. During inference, a significant error in the position prediction of a patch indicates that an anomaly is present within that patch, which was not present in the training distribution. A detailed overview of our method is shown in Figure 1. We evaluate our method on the head CT dataset presented by (Chilamkurthy et al., 2018), which contains images of patients with intracranial haemorrhages and cranial fractures.

Coordinate regression problems have been explored for point-of-interest localization (Nibali et al., 2018) or to propose bounding boxes in object detection (Girshick et al., 2014; Ren et al., 2015). However, these methods focus on finding the location of particular objects within the input image. Compared to this, our method predicts the coordinates of the input patch with respect to the remaining part of the source image. (Lei et al., 2021) proposed a method of estimating positions of patches to locate specific organs or other anatomical structures within whole-body CT scans. In contrast to our method, they simultaneously feed two patches into their network and perform a regression over the relative position of the two patches.

## 2. Method

The method we propose is based on position regression. From an input volume image $I \in \mathbb{R}^{N \times N \times N}$ we extract patches $p_{\mathbf{x}} \in \mathbb{R}^{s_p \times s_p \times s_p}$, with a patch size of $s_p$ at position $\mathbf{x} \in \mathbb{R}^3$ within $I$. The voxels in $I$ can be indexed using three dimensional coordinates $(x, y, z)$ with the coordinates $x, y, z \in \{0, 1, 2, \ldots, N-1\}$. For our purposes we normalize these coordinates to the range $[0, 1]$ such that $\mathbf{x} = (x, y, z) \in [0, 1]^3$.

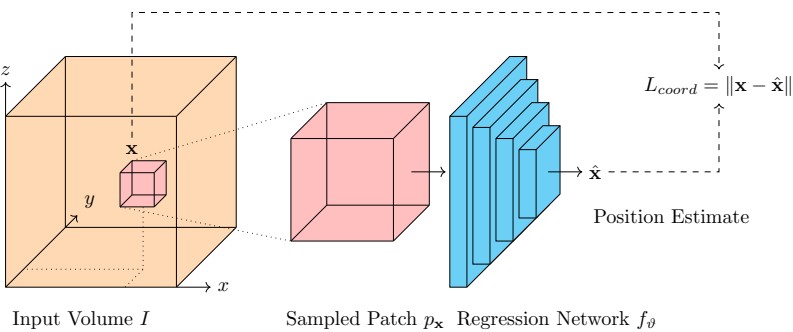

Figure 1: Conceptual overview of the our proposed method.

Using Cartesian coordinates works well for head CT scans, as they always have the same structures in a similar position. For scans of other anatomical structures different that may have different poses, other coordinate systems might be better suited, for instance, a barycentric coordinate system based on some key points.

Given some patch, our network $f_\vartheta : \mathbb{R}^{s_p \times s_p \times s_p} \to \mathbb{R}^3$ with parameters $\vartheta$ is trained to output an estimate $\hat{\mathbf{x}} = f_\vartheta(p_\mathbf{x})$ of the source position $\mathbf{x}$ of the given patch, as shown in Figure 1. Thus we can consider the training of the network as solving a regression problem.

For training, we iterate over our training set: In each iteration, we consider an input volume $I$. We randomly sample coordinates $\mathbf{x}$ from the input volume $I$ and extract the surrounding patch $p_\mathbf{x}$. Then we pass the patch through the network to get the estimated coordinates $\hat{\mathbf{x}} = f_\vartheta(p_\mathbf{x})$ and use the $\ell^2$-norm of the difference between the coordinates of the patch $\mathbf{x}$ and the estimated coordinates $\hat{\mathbf{x}}$ as our training loss

$$L_{coords}(\mathbf{x}, \hat{\mathbf{x}}) = \|\mathbf{x} - \hat{\mathbf{x}}\|.$$

In practice, we use multiple patches in each iteration and average the individual losses $L_{coords}$ to get our training loss that we optimize. These patches are sampled independently from a uniform distribution over input coordinates.

During inference, we compute an output volume $E$ of the same size as the input $I$. For every voxel $I_\mathbf{x}$ at coordinates $\mathbf{x}$ we sample the surrounding patch $p_\mathbf{x}$ of the input $I$, and pass it through the network to get an estimate $\hat{\mathbf{x}}$. Note that the patches centered at the coordinates of two neighbouring voxels will overlap. Then we define the output volume $E$ by computing the reconstruction error for each voxel as $E_\mathbf{x} = L_{coords}(\mathbf{x}, \hat{\mathbf{x}})$.

This allows us to see the output volume $E$ as an error map. Each $E_\mathbf{x}$ shows how well the network $f_\vartheta$ could predict the position of the patch $p_\mathbf{x}$ centered at $\mathbf{x}$. If the value of a voxel in the output volume $E$ is above a certain threshold, the network failed to predict the correct position of the associated input patch. This is the case if the input patch exhibits a structure that is not present at that position in the training images. Therefore, we can check for high values in the output volume $E$ to find regions that appear anomalous.

## 2.1. Architecture

The network we used for the patch position regression (PPR) has a generic image classification network architecture. The detailed structure is displayed in Figure 2. The architecture is parametrized by a network size parameter $m$ to consider a whole range of networks with a different number of parameters. This parameter determines the number of channels in the convolutional blocks as well as the size of the affine layers. We define two blocks, "Downsample" and "Residual" that are used throughout the network. Here "AvgPool" stands for average pooling with a kernel of size 2 and stride of size 2 in every direction. "Conv$(c, k, s)$" stands for a 3D convolution with $c$ output channels, a kernel size of $k$ and a stride of $s$ with spectral normalization. "Linear$(n)$" is an affine transformation with a codomain of dimension $n$.

## 3. Experiments

All runs were trained for 2000 epochs with the Adam optimizer (Kingma and Ba, 2014) with a learning rate lr $= 10^{-4}$. We selected the learning rates empirically based on the training

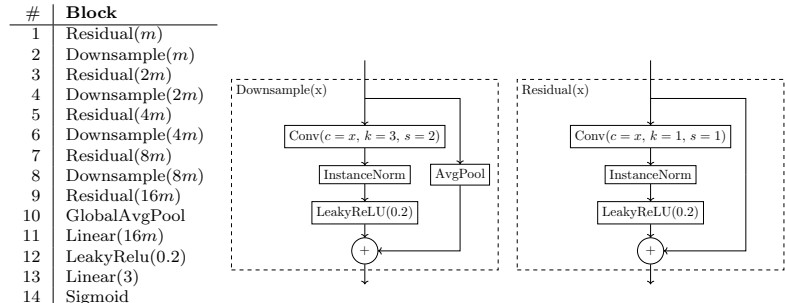

| # | Block |
|---|-------|
| 1 | Residual($m$) |
| 2 | Downsample($m$) |
| 3 | Residual($2m$) |
| 4 | Downsample($2m$) |
| 5 | Residual($4m$) |
| 6 | Downsample($4m$) |
| 7 | Residual($8m$) |
| 8 | Downsample($8m$) |
| 9 | Residual($16m$) |
| 10 | GlobalAvgPool |
| 11 | Linear($16m$) |
| 12 | LeakyRelu(0.2) |
| 13 | Linear(3) |
| 14 | Sigmoid |

Figure 2: Architecture of the Patch Position Regression Network.

loss after 200 epochs. We manually selected a fixed patch size of $s_p = 31$ (i.e. $31 \times 31 \times 31$) voxels for all experiments. (We performed an experiment to examine the influence of the patch size on the performance (Appendix A) and found that in this setting the influence is small.) The size of the patch determines the amount of context: Thus, the amount of information the network gets, but also the sensitivity: There is a trade-off between more context, for a more accurate localization but less sensitivity to anomalies for larger patches and a greater sensitivity but inferior localization accuracy of smaller patches.

### 3.1. Sampling

In each iteration, we sample 256 patches from one volume. This number can be adjusted to the memory budget of the given hardware. We set this number to get a similar run-time as the baseline method (see Section 4.2). Patches that exclusively contain background (i.e. no part of the subject's anatomy) are discarded for the computation of the loss.

### 3.2. Dataset

For training and evaluation we used the public CQ500 dataset (Chilamkurthy et al., 2018), which contains head CT scans. For some patients, there are multiple scans present, for instance with and without contrast enhancement. Three experts determined for each volume whether there is as an intracranial haemorrhage (ICH) for each brain hemisphere and whether a cranial fracture is present. If there was a disagreement between the three raters, we used the majority vote as our ground truth. For our experiments, we used one scan from each patient without contrast enhancement, and discarded all scans that had faulty data (missing slice, wrong anatomical structure etc). The final dataset used in our experiments contains 131 images without anomaly (111 of which are used for training), and 65 images with anomalies. The details of the composition of the dataset are given in Appendix B.

### 3.3. Preprocessing

The volumes are resampled to voxels of size $1 \times 1 \times 1$mm with a total size of $256 \times 256 \times 256$ (that is $N = 256$) voxels. Then each volume is rigidly registered to a manually selected reference volume from the training set using AirLab (Sandkühler et al., 2018). Since CT images have a high dynamic range, we perform a histogram equalization. We segment the

skull and the two brain hemispheres in the images. This segmentation is only used for the evaluation of the method. Furthermore, we separate foreground from background to be able to filter out irrelevant patches during training.

### 3.4. Autoencoder Baseline

We use the basic autoencoder (AE) architecture from (Baur et al., 2021) as a baseline and adapt it to accommodate 3D volumes and to the resolution of the volumes in our dataset. The exact architecture is show in Figure 8 in Appendix C. As a post-processing step, we applied a filter (suggested in (Baur et al., 2021)) of size 5 to the reconstruction error map. We used a median filter for the fractures task, and a grayscale erosion for the ICH task. We optimized over both filter types and multiple kernel sizes to make the comparison as fair as possible.

## 4. Results

For comparison, we trained both our proposed method and the baseline exclusively on *normal* (healthy) data. We used the coordinate reconstruction error to detect anomalies: If the error between the actual and the predicted coordinates is high, we use that as an indication of an anomalous region. The dataset only includes labels of whether an anomaly is present in a given structure (e.g. left hemisphere). For each of these three structures we check whether the error exceeds a certain threshold, in order to predict whether an anomaly is present. Since the performance metrics of the detection, therefore, depends on this threshold, we report the *receiver operating characteristic* (ROC) and the corresponding *area under ROC* (AUROC).

### 4.1. Computational Resources

To evaluate the performance of our method and the baseline method with respect to the computational resources, we trained both methods with various values of the model size parameter $m = 2^0, 2^1, \ldots, 2^6$ to compare the anomaly detection performance to the size of the network. Figure 3 shows the performance as a function of the number of parameters of the networks. We can see that the performance of the networks increases up to some limit, but then decreases again. We conjecture that at a specific size, the training would benefit from more iterations or a better initialization. If we consider the best performing AE networks, we can see that the PPR network requires roughly one (fractures) or two (ICH) orders of magnitude fewer parameters to exceed the performance of the AE. For the remainder, we discuss the best performing PPR models of those that both have fewer parameters *and* used less GPU memory for the training than the best performing AE models. These are marked with an asterisk in Figure 3.

We want to point out though, that even though the memory requirements correlate with the number of parameters, they are also influenced by the actual architecture of the networks as well as the used software frameworks. Furthermore the batch size also influences the memory requirements.

In our experiments, we used batch sizes that would be used for practical purposes, that is we have $\text{bs}_{exp}^{\text{PPR}} = 256$ patches (sampled from one volume) for the PPR models and $\text{bs}_{exp}^{\text{AE}} = 4$

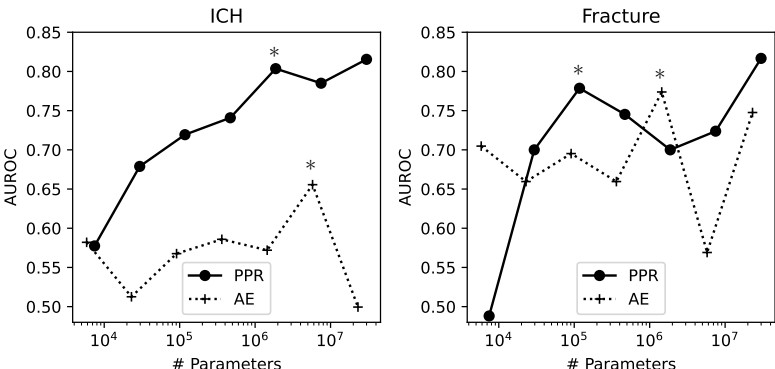

Figure 3: Performance for multiple values of the network size parameter $m = 2^0, 2^1, \ldots, 2^6$. Each of the two plots shows the performance on the test set as a function of the number of network parameters for the two types of anomalies (ICH, Fracture). The asterisk marks the best performing PPR model using fewer parameters and less GPU memory than the best performing AE model (also marked with "$*$").

Table 1: GPU Memory requirements (in MB) during training, given some batch size.

|  | ICH | | Fracture | |
| --- | --- | --- | --- | --- |
| Batch Size | PPR | AE | PPR | AE |
| $\mathrm{bs}_{exp}$ | 4452 | 12548 | 1760 | 7894 |
| 1 | 2194 | 6730 | 1000 | 2742 |

volumes for the AE models. The GPU memory used with these settings for this is shown in the first row of Table 1, and we see that our proposed method uses about a factor of 4 less memory. But even if we only use a batch size of one for the AE models (see second row) the best performing PPR model still uses a factor of about 1.5 less memory. (The memory consumption for all our experiments is shown in Figure 10 in Appendix E.)

It should also be noted that in the ICH experiment, there were PPR models that outperformed the best performing AE model, and used even less memory during training. This illustrates, on the one hand, the general issue of the cost of handling 3D volume data and on the other hand the cost of the image reconstruction branch of the AE models that is not present in the PPR network.

### 4.2. Training and Testing Time requirements

We chose the batch sizes, and number of patches respectively, to result in a similar training time for all models. The time for the AE models was about 26 hours on average, the time for our PPR models is about 22 hours. While the PPR models require less memory for training, they are slower than AE models during inference, which is the price for the patch

based processing: The AE models used needed around 2 seconds to process one volume, while the PPR models needed around 1.5 minutes at the highest resolution.

### 4.3. Qualitative Results

ICH                                      healthy

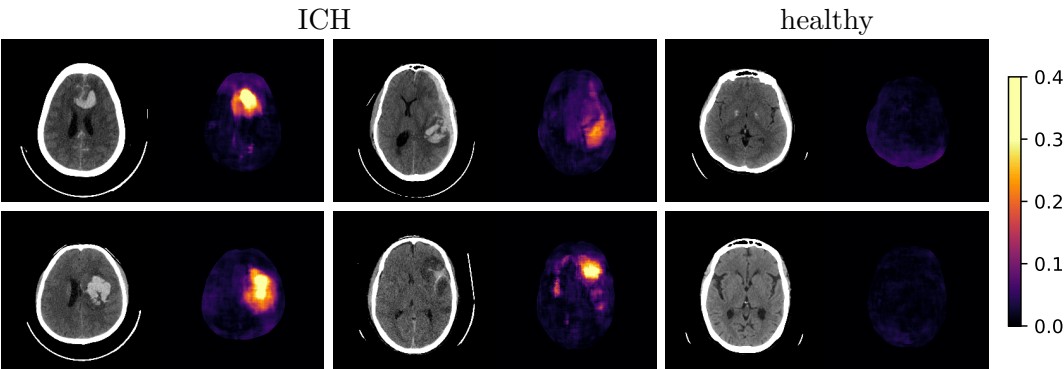

Figure 4: Slices of some selected examples that show the original CT scan with appropriate scaling of the brightness on the left, as well as the error map of our proposed PPR model (with $m = 16$). The images on the left and in the center exhibit an anomaly (ICH) while those on the right are normal (healthy).

We show some slices with examples of ICH in Figure 4 as well as surface renderings of scans of subjects with cranial fractures in Figure 5. (As a reference we also included the same examples for the AE in Appendix D in Figure 9.) It is noticeable that the reconstruction error is high where there is an anomaly. The reconstruction error generally seems to continuously depend on the amount of the patch that is anomalous, as the error maps generally seem to be rather smooth.

The performance for the models used for Figure 4 and 5 are shown in Figure 6. We observe see that the detection of fractures is the more challenging task for our method than the detection of ICH. This might be due to the smaller number of scans available to evaluate it on (see Appendix B). To put these results in context we provided a table with the inter rater agreement on these tasks in Appendix B: The performance in terms of AUROC is around $15 - 16\%$ lower than the average raters.

## 5. Discussion

We have shown that with our proposed approach, we can get a similar performance to AE models with a memory footprint that is up to an order of magnitude lower. This is especially interesting for processing volume data that are common in medical applications, for instance CT- or MRT scans. The lower memory requirements come at a price of a longer inference time. The time needed is still low enough for most applications in the medical field.

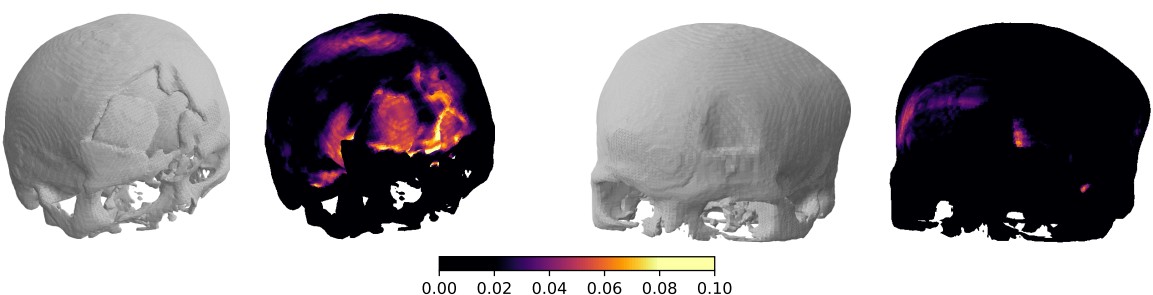

Figure 5: Surface renderings of scans of two subjects with cranial fractures. Both subjects suffer from from fractures of the frontal bone. The left side each shows the scanned part of the skull and the right part shows the same surface coloured according to the error.

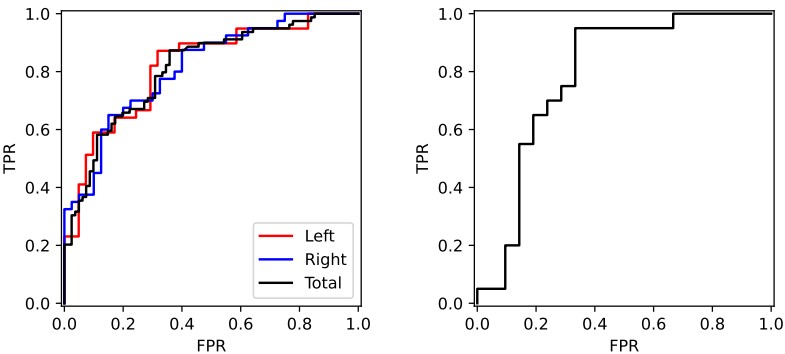

Figure 6: Left: ROC curve for the detection of ICH (AUROC = 0.81), Right: ROC curve for cranial fractures (AUROC = 0.79).

We see the proposed network as a proof-of-concept that would be interesting for further investigation. One of the drawbacks of this method is the limited spatial resolution. This could be addressed with a multi stage coarse-to-fine approach similar to what was proposed in (Lei et al., 2021).

Furthermore, it would be interesting to investigate the influence of the patch size in relation to the spatial extent of the anomalies, and to see if it would be possible to combine it with a tissue classification task. In addition to the patch size it would also be to vary the shape of the patch that is passed into the network

## Acknowledgments

We are grateful for the support of the Novartis FreeNovation initiative and the Uniscientia Foundation (project #147-2018). We would also like to thank the NVIDIA Corporation for the donation of a GPU that was used for our experiments.

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

## Appendix A. Patch Size

To examine the influence of the patch size (see Section 3) we evaluated the ICH task with identical settings but different patch sizes. For each of the patch sizes we trained a model from scratch with the same configuration as in the other experiments. In Figure 7 we show the AUROC score as a function of the patch size for the detection of the haemorrhages in each brain hemisphere (red and blue) as well as the total score for both hemispheres combined. Across all six patch sizes the performance slightly changes by about $\pm 0.06$. Comparing that value with the variation of the individual brain hemispheres we conclude that the influence of the patch size in these experiments in negligible. For other anomalies or modalities this might be different though.

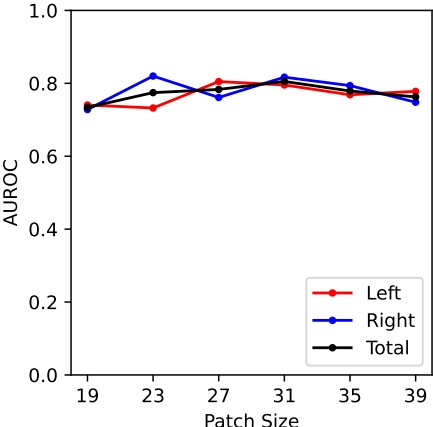

Figure 7: Performance of the PPR network in terms of AUROC as a function of the patch size $s_p = 19, 23, 27, 31, 35, 39$ for the ICH task.

## Appendix B. CQ500 Dataset

The scans with bleeding or a fracture were considered *anomalous*, while the scans without any of these findings were considered *healthy*. The test set contains 86 volumes in total, including 21 from the healthy set (i.e. without anomalies), 20 with a fracture, 39 with

bleeding in the left hemisphere and 47 with bleeding in the right hemisphere. Note that these sets are not disjoint, Table 2 shows the actual distribution. Our test set consists of the anomalous scans as well as 21 randomly chosen scans from the healthy set. The remaining scans of the healthy set were used for training.

Table 2: Distribution of the anomalous data in the test set. Overlined column/row names indicate the absence of the given feature. Each entry shows the number of volumes with that combination of the presence/absence of these three features.

| | Bleeding Left | | $\overline{\text{Bleeding Left}}$ | |
|---|---|---|---|---|
| | Bleeding Right | $\overline{\text{Bleeding Right}}$ | Bleeding Right | $\overline{\text{Bleeding Right}}$ |
| Fracture | 4 | 3 | 7 | 6 |
| $\overline{\text{Fracture}}$ | 16 | 16 | 13 | 21 |

To characterize the variability of the ground truth within the three raters, we compute the Fleiss-Kappa as well as the pairwise AUROC in Table 3.

Table 3: Inter rater agreement expressed using the Fleiss-Kappa as well as AUROC for each rater compared to the majority vote for each feature.

| | $\kappa$ | AUROC | | | |
|---|---|---|---|---|---|
| | | R1 | R2 | R2 | AVG |
| Bleeding Left | 0.745 | 0.877 | 0.982 | 0.985 | 0.948 |
| Bleeding Right | 0.705 | 0.893 | 0.945 | 0.964 | 0.934 |
| Fracture | 0.632 | 0.901 | 0.955 | 0.921 | 0.926 |

## Appendix C. Autoencoder Baseline

We trained the AE network for 2000 epochs with the Adam optimizer with lr = 0.001. The architecture of the AE is shown in Figure 8: The architecture is parametrized by $m$ to be able to consider networks of different sizes. We define two blocks, "Downsample" and "Upsample", that are used throughout the network. "[Transp]Conv$(c, k, s)$" stands for a 3D [transposed] convolution with $c$ output channels, a kernel size of $k$ and a stride of $s$. The AE models were trained using a pixel-wise $L_1$-loss.

| # | Block |
|---|-------|
| 1 | Downsample($m$) |
| 2 | Downsample($2m$) |
| 3 | Downsample($4m$) |
| 4 | Downsample($8m$) |
| 5 | Downsample($16m$) |
| 6 | Upsample($8m$) |
| 7 | Upsample($4m$) |
| 8 | Upsample($2m$) |
| 9 | TranspConv($c = 1, k = 4, s = 2$) |
| 10 | Sigmoid |

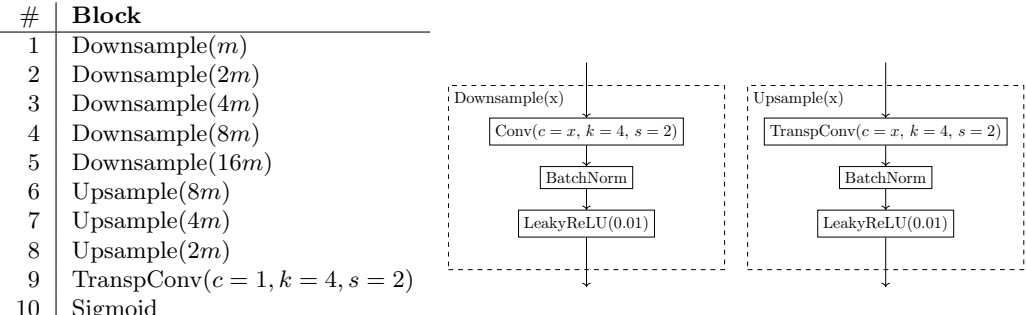

Figure 8: Architecture of the AE network.

## Appendix D. Qualitative Results AE

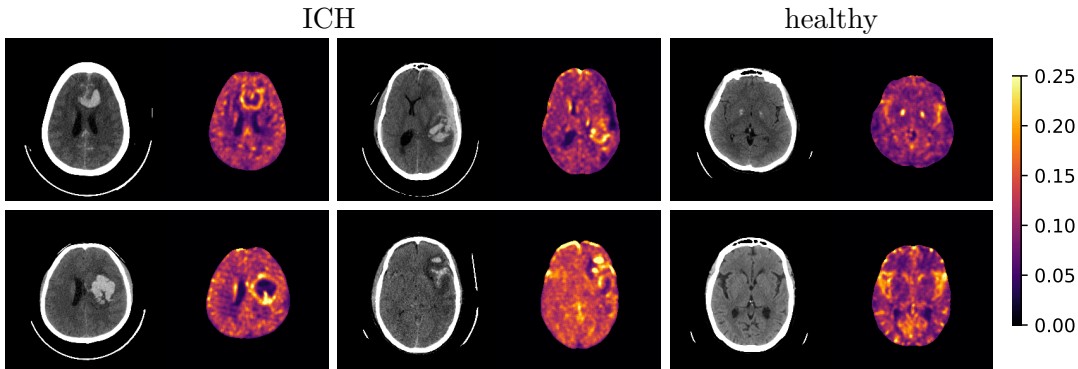

Figure 9: Slices of some selected examples that show the original CT scan with appropriate scaling of the brightness on the left, as well as the error map the baseline AE method. The images on the left and in the center exhibit an anomaly (ICH) while those on the right are normal (healthy).

## Appendix E. Memory Consumption

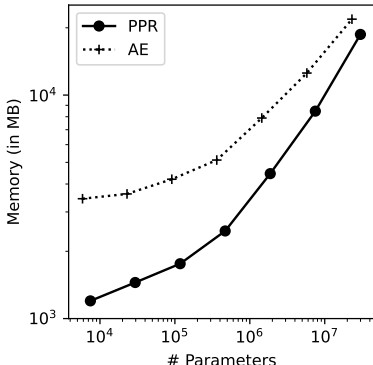

Figure 10: GPU Memory consumption as a function of the number of parameters for the AE and PPR with network size parameter $m = 2^0, 2^1, \ldots, 2^6$ and batch sizes $\text{bs}_{exp}$ as used in the experiments.

