# OpenReview forum: "Position Regression for Unsupervised Anomaly Detection"
_MIDL.io/2022/Conference — MIDL 2022_

### Official Review · Reviewer_ojhS · 2022-01-18

**Confidence:** 4
**Preliminary Rating:** 3
**Recommendation:** Poster

**Summary:**

The authors propose a position regression approach for unsupervised anomaly detection. This approach is patch-based, thus, it requires much less computational resources during the training. The authors assume that, if trained on healthy data, the proposed patch position regression (PPR) network, would yield distant coordinates for anomalous patches during the inference, while the position of healthy patches can be predicted correctly.  In their experiments, the authors show the suitability of the approach for pathologies present in brain CTs and compare the memory requirement for the PPR vs. a regular autoencoder.

**Strengths:**

- The paper is written clearly and is easy to follow and understand.
- The idea for unsupervised detection using patch position regression is interesting. The proposed method shows advantages including a more simple architecture and thus less parameters and required memory.
- The quantitative and qualitative results look promising and can be accepted as proof-of-concept
- A public dataset is used and the code will be made available on Github


**Weaknesses:**

- The quantitative evaluation of the method is not very strong. The authors do show the ROCs and the resulting AUCs, however, this is the only evaluation in terms of accuracy of the pathology detection. Also, the authors do not directly compare to other methods (e.g. autoencoder) in terms of AUCs or qualitative results.
- The baseline AE method is not described well enough. It is not clear what AE the authors use since they reference to Baur et al., however, this is a comparison paper which evaluates many different autoencoders. Fig. 3 shows results of the AE in relation to the number of parameters required, however, those results are not further numerically explained (it is hard to read the concrete numbers in the figure).
- Comparing a patch-based method to a non-patch-based method in terms of memory requirements might be unfair. A patch-based AE might be the fairer comparison here.
- Using Cartesian coordinates works good for the chosen data. The authors do mention that other coordinate systems might be better suited for other data, however, this approach will probably be very hard to apply to data like whole-body CT/MRI even after prior registration.

**Deanonymize Review:**

no

**Detailed Comments:**

Further remarks:
- A more in-depth discussion on the choice of the patch-size would be nice. Maybe even an experiment. My intuition is that the size of the patches depends on the approximate pathology size and image size.
- A colorbar in Fig. 5 would be nice, also the visibility in this figure is not very good.



**Final Rating After The Rebuttal:**

4: Weak Accept

**Justification Of The Final Rating:**

I thank the authors for their thorough review. The new included experiments and also the added justifications add to the value of the work. Overall an interesting approach, so I change my rating to weak accept.

**Paper Type:**

both

**Questions To Address In The Rebuttal:**

- A concrete comparison between the AE and the proposed method would make the results more convincing.
- More explanation of the AE approach and discussion on the patch size
- Additional experiments on patch size and other comparison methods (maybe patch-based) would highly improve the work

**Special Issue:**

no

---

### Official Review · Reviewer_qBri · 2022-01-24

**Confidence:** 4
**Preliminary Rating:** 4
**Recommendation:** Poster

**Summary:**

The authors present a novel method for unsupervised anomaly detection. The method is trained on data from only healthy patients and predicts the location of extracted patches within a volume. The expectation is that when the trained model is applied to patches belonging to an anomalous region it will have a higher error trying to predict the position of the patch, compared to a healthy region. By using each voxel as the center of a patch, an error value can be assigned to each voxel, thereby creating an error map where the higher values correspond to anomalies. Unlike state-of-the-art approaches, the presented method does not reconstruct the whole image volume and achieves comparable performance while requiring less memory on the dataset tested.

**Strengths:**

Unsupervised anomaly detection is one of the biggest challenges in medical imaging. The authors address this problem with a novel approach and show that their method achieves comparable results to a well established autoencoder (AE) method, while requiring less memory during training. The paper provides a concise and understandable overview on the field of unsupervised anomaly detection and it becomes clear how the proposed method differs from established methods. State-of-the-art unsupervised anomaly detection methods learn to reconstruct the entire image volume from a low-dimensional representation of the input data. In contrast, the proposed method determines only the position of extracted patches. This results in lower memory requirements during training. The proof-of-concept work of the authors provides a basis for further investigation into more energy efficient and scalable methods for unsupervised anomaly detection.

**Weaknesses:**

It is understandable that this method is a proof of concept, but there are still several points that need to be mentioned:
- While the method used less memory during training, it appears slow during inference (1.5 minutes vs 2 seconds for the competing method). This might hinder potential adoption, especially if the problem scales to larger resolutions or datasets.
- The study didn’t compare to the actual state-of-the-art methods that involve high dimensional image reconstruction, which currently seems to be VAE (Bauer et al 2020) or maybe a GAN-based method. Although, that might not necessarily be a problem with a proof of concept paper, where an AE might have been more simple to provide a benchmark. Further points for investigation could also be the comparison with other architectures for position regression, for example a transformer based architecture.
- The study data contained ground truth labels on whether there was an anomaly on the left side, the right side, or the skull. As a result, the method’s ability to localize is not thoroughly quantified.
- The connection between the amount of parameters and the memory requirements is not given.
- The discussion is very small, especially for a novel method. The authors could have provided more information on the advantages and limitations of the method, as well as more potential next steps and other relevant applications. For the current task, is a long inference time more preferable to lower memory consumption during training and why?


**Deanonymize Review:**

yes

**Detailed Comments:**

- The results section seems to be not very well organized. The main results are under “Computational resources”. Time comparison should probably be right after memory comparison and not in the end.
- In the second paragraph of introduction, there should be some citation(s) regarding the most common methods belonging to the first type (some are provided afterwards, but as it stands it is less helpful to a potential reader).
- The language can be improved in various places.


**Paper Type:**

methodological development

**Questions To Address In The Rebuttal:**

- What new possibilities arise with respect to clinical applications from the suggested method?
- What is better for possible applications, less memory requirements or fast inference?
- What happens with a larger network size for both network types (if that is feasible)? Does the autoencoder outperform the PPR method on more hyperparameters with respect to memory requirements?
- What behavior do you expect from other network architectures performing position regression?


**Special Issue:**

no

---

### Official Review · Reviewer_TPQg · 2022-01-24

**Confidence:** 5
**Preliminary Rating:** 2
**Recommendation:** Poster

**Summary:**

This paper proposes an unsupervised anomaly detection method based on a CNN model trained to regress spatial coordinates of patches extracted from healthy subject 3D image volume. This method is applied on the public CQ500 dataset which contains head CT scans with intracranial haemorrhages or cranial fractures. Performance results are reported as the ability of the model to discriminate normal and pathological images (intracranial fracture, left or right bleeding).



**Strengths:**

-Research in the domain of unsupervised anomaly detection is a currently very active in the community.
-The proposed idea to regress the patch localization is original with regards to the main publication stream of models based on the reconstruction errors of autoencoders trained on normal population.
-The paper is clear and well written


**Weaknesses:**

-Performance of the proposed model is evaluated on a quite simple classification task at the image level, that is discriminating patient images which contains visible lesions (as depicted on fig 4) from normal looking images of healthy subjects:
-Regarding this task, I am wondering if a simple thresholding of the CT scans might not provide the same range of AUC results, at least regarding the intracranial haemorrhage, as recently performed by Meissen et al (MICCAI 2021 BrainLes Workshop) on another dataset?
-Error maps reported on fig 4 are good looking which is encouraging. The method might indeed perform well in more complex tasks such as segmentation. The authors should consider such more challenging tasks to demonstrate the soundness of their method.
-It would be interesting to show error maps achieved by the VAE model, as well, in order to compare them with those achieved with the proposed method and correlate these error maps to the classification performance task (VAE is indeed shown to perform poorly).


**Deanonymize Review:**

no

**Detailed Comments:**

-The model is based on regressing patch localization within a 3D image volume, thus assuming that patches extracted at the same location in the series of healthy training data should look almost similar, thus requiring that all image data are coregistered. It would be interesting to evaluate the impact of registration accuracy on the error maps.
-Please provide details on the post-processing of the error maps. AUC was plotted by varying the threshold applied to the error maps. How were the resulting binary maps processed? Did the authors perform any clustering or apply any rules to consider a true positive detection. Please add some details.


**Final Rating After The Rebuttal:**

3: Borderline

**Justification Of The Final Rating:**

I thank the authors for their thorough review. I agree to upgrade my rating to ‘borderline’, because the proposed idea is original. Still more experiments are required to demonstrate it does not face the same limitation as unsupervised  anomaly detection models based on reconstruction errors.

**Paper Type:**

both

**Questions To Address In The Rebuttal:**

The authors should address comments reported in the two previous sections, especially regarding comparison with VAE models. They also should provide more details on the processing (registration etc..) of the error maps and on the performance analysis and discuss the limitation of the proposed method.

**Special Issue:**

no

---

### Meta-Review · Area_Chair_cU5G · 2022-02-14

**Recommendation:** Accept (Poster)
**Confidence:** 5

**Metareview:**

The paper is well written and presents a creative and original approach to anomaly detection in head CT images. A limitation that the reviewers point out is the requirement of a fixed coordinate system across images, and hence the necessity of accurate registration. Nevertheless, this is an interesting methodological contribution to MIDL that might be further developed in the future.

---

### Decision · Program_Chairs · 2022-02-28

Accept